# Two Amino Acid Substitutions Improve the Pharmacological Profile of the Snake Venom Peptide Mambalgin

**DOI:** 10.3390/toxins17030101

**Published:** 2025-02-21

**Authors:** Dmitry I. Osmakov, Timur A. Khasanov, Ekaterina E. Maleeva, Vladimir M. Pavlov, Victor A. Palikov, Olga A. Belozerova, Sergey G. Koshelev, Yuliya V. Korolkova, Igor A. Dyachenko, Sergey A. Kozlov, Yaroslav A. Andreev

**Affiliations:** 1Shemyakin—Ovchinnikov Institute of Bioorganic Chemistry, Russian Academy of Sciences, Ul. Miklukho-Maklaya 16/10, 117997 Moscow, Russia; hasanov.ta@phystech.edu (T.A.K.); katerina@1ns.ru (E.E.M.); o.belozyorova@gmail.com (O.A.B.); sknew@yandex.ru (S.G.K.); july@ibch.ru (Y.V.K.); serg@ibch.ru (S.A.K.); aya@ibch.ru (Y.A.A.); 2Moscow Center for Advanced Studies, Kulakova Str. 20, 123592 Moscow, Russia; 3Branch of the Shemyakin-Ovchinnikov Institute of Bioorganic Chemistry, Russian Academy of Sciences, 6 Nauki Avenue, 142290 Pushchino, Russia; v.m.pavlov29@gmail.com (V.M.P.); vpalikov@bibch.ru (V.A.P.); dyachenko@bibch.ru (I.A.D.)

**Keywords:** mutagenesis, ASIC channels, electrophysiology, peptide, analgesic effect

## Abstract

Mambalgins are peptide inhibitors of acid-sensing ion channels type 1 (ASIC1) with potent analgesic effects in models of inflammatory and neuropathic pain. To optimize recombinant peptide production and enhance pharmacological properties, we developed a mutant analog of mambalgin-1 (Mamb) through molecular modeling and site-directed mutagenesis. The resulting peptide, Mamb-AL, features methionine-to-alanine and methionine-to-leucine substitutions, allowing for a more efficient recombinant production protocol in *E. coli*. Electrophysiological experiments demonstrated that Mamb-AL exhibits three-fold and five-fold greater inhibition of homomeric ASIC1a and ASIC1b channels, respectively, and a two-fold increase in inhibition of heteromeric ASIC1a/3 channels compared with Mamb. In a mouse model of acetic acid-induced writhing pain, Mamb-AL showed a trend toward stronger analgesic efficacy than the wild-type peptide. These improvements in both production efficiency and pharmacological properties make Mamb-AL a valuable tool for studying ASIC channels and a promising candidate for analgesic drug development.

## 1. Introduction

Acid-sensing ion channels (ASICs) belong to the epithelial sodium channel/degenerin superfamily of ion channels [1]. A key function of ASICs is their ability to respond to decreases in extracellular pH [2]. In mammals, four genes encode six isoforms of ASICs: ASIC1a, ASIC1b, ASIC2a, ASIC2b, ASIC3, and ASIC4 [3]. Structural insights have been obtained for only one isoform, ASIC1, through X-ray crystallography and cryo-electron (cryo-EM) microscopy studies [4,5,6,7]. ASICs function as trimeric complexes, where each subunit consists of short intracellular *N*- and *C*-termini, two transmembrane domains, and an extracellular cysteine-rich domain [4]. The extracellular domain features six distinct regions, commonly referred to as the “wrist”, “palm”, “finger”, “knuckle”, “β-ball”, and “thumb” [4,7].

ASICs are abundantly expressed in neurons of both the peripheral and central nervous systems [8]. These channels have been implicated in a variety of physiological processes, including synaptic plasticity, learning, memory, fear, and anxiety [9,10,11,12]. Moreover, they play roles in pathological conditions such as neurodegeneration following ischemia and the perception of acid-induced, inflammatory, and postoperative pain [13,14,15,16]. Given their functional diversity, ASICs have emerged as attractive pharmacological targets for the treatment of ischemic and neurodegenerative diseases, bladder disorders, and inflammatory and neuropathic pain [16,17,18].

Among the ligands of ASICs, peptide-based ligands are of particular interest due to their high potency and specificity [19,20]. Notable examples include the mambalgins (Mamb-1, -2, -3), which are peptides isolated from snake venom and differ from each other by a single amino acid residue [20]. All three isoforms of mambalgins (Mamb), which exhibit the same pharmacological properties, inhibit ASIC1 channels (both in homo- and heteromeric forms) at nanomolar concentrations by stabilizing the channel in its closed state [20,21,22]. Structurally, Mamb belongs to the family of three-finger toxins and consists of 57 amino acid residues [21]. Using cryo-EM microscopy, the structure of the Mamb-1 and ASIC1 complex revealed that fingers I and II of the peptide interact with the thumb domain of the channel [23,24]. This finding was subsequently confirmed through functional studies involving peptide and channel mutants [25,26,27]. In various in vivo tests, mambalgins have demonstrated potent analgesic effects. For instance, in models of inflammatory pain, these peptides alleviated thermal and mechanical hypersensitivity, while in a migraine model, they reduced mechanical allodynia [21,28,29]. Despite their pharmacological significance, obtaining mambalgins poses several challenges: (a) natural source extraction, limited availability of venom, and the complexity of isolating peptides from a multicomponent mixture [21]; (b) chemical synthesis, high costs associated with synthesizing long peptides, and the need for proper refolding [26,30,31]; (c) recombinant production in bacterial expression systems: low yields due to the requirement for refolding [32,33].

In this study, we developed a mutant analog of Mamb-1, the peptide Mamb-AL, which outperforms the original molecule in several key aspects. First, we established a more efficient recombinant production strategy for Mamb-AL, achieving a higher yield. Second, electrophysiological experiments on ASIC1 channels demonstrated that Mamb-AL exhibits several-fold greater inhibitory potency compared with Mamb. Consequently, in an acid-induced pain test in mice, Mamb-AL showed a trend toward a stronger analgesic effect than the original peptide.

## 2. Results

### 2.1. In the Interaction Models, the Mutant Analog of Mambalgin Exhibits Tighter Binding to the ASIC1a Channel than the Wild-Type Peptide

In our study, we hypothesized that replacing Met residues in the mambalgin (Mamb-1) could enhance its functional properties. We based our approach on earlier findings [26,27], which showed that substituting Met16 and Met25 in Mamb-1 or Met16 in Mamb-3 with Ala residues, while not significantly altering activity, demonstrated a tendency to improve peptide action on ASIC1a. To test this hypothesis, we generated a mutant analog (Mamb-AL) of Mamb-1, in which Met16 and Met25 residues were substituted with Ala and Leu, respectively (Figure 1A). Notably, physicochemical properties such as isoelectric point, extinction coefficient, and hydrophobicity (GRAVY score), calculated based on the primary structure of these peptides [34], remained unchanged between Mamb-1 and Mamb-AL (Table A1). Moreover, Mamb-AL exhibited a higher aliphatic index than the wild type, which may indicate greater stability.

Next, we performed molecular modeling of the interactions between Mamb-1, Mamb-AL, and the model structure of the rat ASIC1a channel (rASIC1a). The results revealed that Mamb-AL forms tighter contacts with the channel compared with wild-type Mamb (Figure 1B,C).

In our analysis of intermolecular interactions, the peptides did not show significant differences in their interaction or total scores with the channel (Table A2). Therefore, we focused on peptide residues known to play critical roles in rASIC1a binding. As shown in Figure 1B, residues His6, Phe27, Arg28, and Leu34, which make some of the most significant contributions to peptide activity, were positioned statistically closer to their interacting partners on the channel in Mamb-AL than in Mamb. Notably, for His6 and Phe27, the reduction in distance exceeded 1 Å. Thus, based on the modeling results, Mamb-AL appears to be a more effective ligand for rASIC1a than wild-type Mamb.

### 2.2. High-Yield Production of the Mutant Peptide Mamb-AL

Mamb peptides are relatively large (57 amino acids), making their chemical synthesis not only expensive but also requiring a refolding step [25,26,30]. An alternative approach involves producing recombinant analogs in bacterial expression systems, followed by purification either from inclusion bodies or the periplasmic space. In the case of the inclusion bodies technique, refolding was a critical step, resulting in a low yield of the final product, approximately 0.2–0.5 mg per gram of bacterial biomass [32,33]. Periplasmic expression yielded similarly low amounts of peptide (∼0.2 mg/L) [22]. In this study, we designed a genetic construct for the production of a chimeric protein, in which the target peptide was fused to thioredoxin—a well-characterized thiol antioxidant known to facilitate the folding of cysteine-rich peptides [35,36]. We then optimized a protocol for its isolation in sufficient quantities. The chimeric protein was expressed in the cytoplasm of bacterial cells and purified using metal-affinity chromatography. It was subsequently cleaved with cyanogen bromide (CNBr), and the target peptide was separated from thioredoxin fragments via reversed-phase HPLC on a Jupiter C5 column (Figure 2A). It is worth noting that under these conditions, the retention time of Mamb-AL was 32.6 min, while that of the wild-type peptide was 33.5 min. The purified Mamb-AL peptide (purity >97% (Figure A1)) was isolated using a Synergi Polar-RP column with a different stationary phase (Figure 2B).

The molecular mass of the target peptide, measured by electrospray ionization mass spectrometry, was 6476.12 Da (Figure 2C), which closely matched the theoretically calculated mass of 6476.42 Da. Thus, this approach (detailed in the Materials and Methods section) allowed us to achieve a yield of Mamb-AL of approximately 2.75 mg/L (or 2.5 mg per gram of bacterial biomass), which is an order of magnitude higher than the yields reported in previous studies for Mamb.

### 2.3. Mamb-AL Is a Stronger Negative Modulator of rASIC1a than Wild-Type Mamb

The functional activity of Mamb-AL and its comparison with wild-type Mamb were assessed using two-electrode voltage clamp (TEVC) recordings in *Xenopus laevis* oocytes heterologously expressing rASIC1a. When comparing the inhibitory effects of the two peptides on homomeric rASIC1a, it was observed that at a pH stimulation of 5.5, the half-maximal inhibitory concentration (IC_50_) values for Mamb and Mamb-AL were 110 ± 12 nM and 62 ± 9 nM, respectively (Figure 3A,B). Notably, the inhibition was not complete, with a maximum effect of approximately 90%.

At a weaker stimulus of pH 6.7, the difference in IC_50_ values between the two peptides became even more pronounced, with Mamb and Mamb-AL showing IC_50_ values of 86 ± 10 nM and 25 ± 3 nM, respectively (Figure 3C,D). Under these conditions, both peptides achieved complete inhibition (100%). Thus, across different stimulation strengths (pH 5.5 and 6.7), Mamb-AL demonstrated significantly stronger inhibitory effects. At pH 6.7, Mamb-AL outperformed Mamb by more than threefold (Figure 3E). Mamb-AL also showed a superior effect in accelerating the desensitization rate of rASIC1a. At pH 5.5-induced currents, Mamb-AL increased the desensitization rate by 45%, compared with only a 24% increase observed for Mamb (Figure 3F).

### 2.4. Mamb-AL Has a Stronger Effect on Other ASIC1-Containing Channels than Mamb

Mambalgins also exhibit inhibitory effects on the ASIC1b isoform, a splice variant of ASIC1a that is predominantly expressed in the peripheral nervous system [21,37]. Investigation of peptide activity on rat ASIC1b (rASIC1b) showed that Mamb and Mamb-AL inhibited the channel with IC_50_ values of 119 ± 3 nM and 22 ± 1 nM, respectively (Figure 4A,B). Thus, Mamb-AL displayed five times greater potency against rASIC1b compared with Mamb.

Moreover, the IC_50_ value of Mamb-AL for rASIC1b surpassed those of all three mambalgins, including Mamb-1 (IC_50_ ranging from 39 nM [27] to 192 nM [21]), Mamb-2 (IC_50_ of 103 nM [25]) and Mamb-3 (IC_50_ ranging from 33 nM [22] to 50 nM [27]). Mamb and Mamb-AL also partially inhibited heteromeric rat ASIC1a/3 channels (rASIC1a/3) (Figure 4C). Here, Mamb-AL again significantly outperformed the wild-type peptide, inhibiting rASIC1a/3 by 32% at a concentration of 100 nM, compared with only 13% inhibition by Mamb (Figure 4D). Interestingly, it has been reported that Mamb-1 has no effect on this channel [21]. For rat ASIC3 channels, neither wild-type Mamb nor Mamb-AL exhibited any detectable effects (Figure 4E,F).

### 2.5. Mamb and Mamb-AL Exhibit Analgesic Effect

ASIC channels play a critical role in the perception of acid-induced pain [8,38]. As effective inhibitors of ASIC1-containing channels, Mamb and Mamb-AL could serve as potential therapeutics for this type of pathology. To evaluate this hypothesis, we employed the acetic acid-induced writhing test as an in vivo model of acute pain. In this model, intraperitoneal administration of 0.6% acetic acid provokes a stereotypical behavior in mice characterized by abdominal contractions (acetic acid-induced writhes). Pretreatment with Mamb or Mamb-AL (0.01 and 0.1 mg/kg) 90 min prior to testing significantly reduced the number of writhes (Figure 5).

The number of writhes in the control group treated with saline was 41 ± 8. Administration of wild-type peptide at doses of 0.01 and 0.1 mg/kg decreased the number of writhes by approximately 29% and 27%, respectively. In contrast, Mamb-AL showed a stronger analgesic effect, reducing the number of writhes by ~41% and ~34% at the same doses, respectively. Thus, for the first time, the analgesic effect of mambalgins was demonstrated in a model of acute acid-induced pain. Furthermore, Mamb-AL, as a more potent inhibitor of ASIC1-containing channels, showed a trend toward a stronger analgesic effect than the wild-type Mamb in terms of pain relief.

## 3. Discussion

Mambalgins (Mamb) are well-studied peptide ligands of ASIC1 channels, with both their pharmacophores and binding sites on the channel well characterized [23,25,26,27,30,31]. However, previous studies focusing on mambalgin mutagenesis, coupled with the inherent challenges of peptide production, have consistently demonstrated only reduced activity of these mutants on the channel [24,26,27,31]. The goal of our work was to design a mambalgin analog that would not only simplify the production process but also enhance the pharmacological properties of the original molecule.

Building on earlier findings [26,27], which showed that substituting each methionine residue individually led to an enhancement, albeit not statistically significant, of Mamb’s action on ASIC1a, we designed a mutant analog of Mamb-1, named Mamb-AL. In this variant, Met16 and Met25 were replaced with Ala and Leu, respectively, while theirphysicochemical properties remained largely similar to those of the parent peptide. Molecular modeling of the interactions between wild-type Mamb-1, the mutant Mamb-AL, and the rat ASIC1a channel (rASIC1a) revealed no significant differences in their binding energies. However, notable differences were observed in the interaction distances between residues critical for peptide activity and their corresponding sites on the channel—Mamb-AL exhibited tighter contacts than Mamb-1. Based on this, we hypothesized that these structural differences could influence the peptide’s activity and, therefore, proceeded with its experimental characterization.

Previous studies faced significant challenges in obtaining sufficient amounts of Mamb peptides due to several factors: (a) the limited availability of the natural source (i.e., snake venom) from which the peptide could be isolated [21,28]; (b) the relatively large size of the peptide (57 amino acids), making chemical synthesis followed by refolding an expensive approach [25,26,30]; (c) the low yields of recombinant analogs expressed in bacterial systems, followed by purification from the periplasmic space [22,27]; and (d) the complex procedure for isolating recombinant peptides from bacterial inclusion bodies, which involved optimizing refolding conditions, dialysis, ion exchange and size-exclusion chromatography—ultimately yielding low amounts of the target peptide [32,33]. In contrast, we developed a recombinant expression strategy utilizing thioredoxin fusion, which enabled the production of the target peptide in the cytoplasm of *E. coli* in a soluble form. This approach allowed for efficient purification in just two steps—affinity chromatography followed by reverse-phase chromatography after CNBR cleavage. The resulting high yield of Mamb-AL, significantly exceeding previously reported yields of Mamb, made it possible to produce sufficient quantities for further biological studies.

Mamb peptides belong to the three-finger toxin (3FTx) family, a diverse group of snake venom peptides. While many 3FTx members are known for their cytolytic activity [39], a significant subset does not induce cell lysis but instead exerts specific effects on various receptors and ion channels [40]. Mamb peptides fall into this latter category. Extensive in vitro and ex vivo studies using COS-7, CHO, and human neuroblastoma cells [21,41], as well as oocytes [26,42], mouse hippocampal and dorsal spinal cord neurons [21], and human stem cell-derived sensory neurons [43], have demonstrated that Mamb peptides exhibit no signs of cytotoxicity. Furthermore, our in silico analysis did not indicate any potential hemolytic activity for either Mamb or Mamb-AL. Using HemoPI (Hemolytic Activity Prediction), the PROB scores for peptide fragments scanned from Mamb and Mamb-AL ranged from 0.47 to 0.51, where a score of 1 indicates a high likelihood of hemolytic activity, and 0 suggests a very low likelihood [44].

Moreover, in various in vivo models of acute, chronic inflammatory, and neuropathic pain, Mamb peptides consistently exhibited potent analgesic effects without any apparent side effects, regardless of the route of administration (intrathecal, intravenous, or intraplantar) [21,28,29]. Elegant experiments have demonstrated that the predominantly analgesic effect of Mamb is mediated by its inhibitory action on the ASIC1b isoform. This makes our findings particularly intriguing, as we observed that after the Mamb mutation, its inhibitory activity on ASIC1-containing channels was enhanced. This suggests that the mutation may have increased the peptide’s affinity for ASIC, allowing it to exert inhibitory effects at lower concentrations. However, unlike the wild-type peptide, Mamb-AL was unable to fully block channel activity, leading to a reduced maximum inhibition rate. Despite this, the Hill coefficients for both peptides indicated positive synergy (1.54 ± 0.03 for Mamb and 1.4 ± 0.1 for Mamb-AL). Similar to what has been observed for mambalgin-3 [27], it is likely that these mutations alter the peptide’s binding site, thereby modifying its interaction with ASIC channels.

In in vivo mouse experiments, we demonstrated for the first time that both Mamb and Mamb-AL produce significant analgesic effects in the acetic acid-induced writhing pain model. Notably, both peptides exhibited their strongest effects at a relatively low dose of 0.01 mg/kg. However, when the dose was increased tenfold (0.1 mg/kg), the effect did not intensify; instead, there was a statistically non-significant trend toward a weaker response. This phenomenon, known as hormesis, is well-documented for biologically active compounds [45], including negative modulators of ASIC channels [46].

## 4. Conclusions

In this study, we developed Mamb-AL, a novel analog of the analgesic peptide mambalgin, with improved pharmacological properties. By optimizing the recombinant production protocol, we significantly increased peptide yield while eliminating the need for inclusion body isolation and refolding. Functionally, Mamb-AL exhibited stronger inhibitory activity against ASIC1a and demonstrated superior efficacy on the ASIC1b channel, a key player in peripheral pain perception. In an in vivo pain model, Mamb-AL provided a significant analgesic effect at relatively low systemic doses (0.01–0.1 mg/kg), showing a trend toward surpassing the efficacy of the original molecule. These findings highlight Mamb-AL as a valuable tool for ASIC channel researchand a promising candidate for acid-induced pain treatment.

## 5. Materials and Methods

### 5.1. Materials

The peptide Mamb (mambalgin-2) was produced as a recombinant analog in *Escherichia coli*, following the protocol described by Bychkov et al. [32]. The salts used for buffer preparation were obtained from Sigma-Aldrich (St. Louis, MO, USA) and ISOLAB Laborgeräte (Eschau, Bavaria, Germany). Collagenase type I was obtained from Worthington Biochemical Corporation (Lakewood, NJ, USA). Gentamicin and penicillin were purchased from neoFroxx GmbH (Einhausen, Germany).

### 5.2. Molecular Docking Analysis of Peptide Interactions with the Rat ASIC1a

We performed molecular docking of peptides with the rat channel ASIC1a as described in [47]. Mutant analogue of Mamb (PDB ID: 7ULB) was generated using PyMOL v.3.0 (Schrödinger LLC, New York, NY, USA). Initial structural relaxation of Mamb and its mutant variant, Mamb-AL, was carried out using the Rosetta Relax protocol to prepare the models. Subsequently, molecular docking was performed with RosettaDock 5.0 [48]. We analyzed the resulting models to evaluate the interactions between the peptides and the ASIC1a channel, with a particular focus on the amino acid residues critical for binding.

### 5.3. Mamb-AL Gene Synthesis

The DNA encoding the Mamb-AL mature peptide sequence was constructed using synthetic oligonucleotides and PCR amplification. The target PCR fragment was amplified with a set of seven primers: forward primer 1 containing a restriction enzyme EcoRI site and the Met codon for cyanogen bromide (CNBr) cleavage (5′-AGATGAATTCAATGCTGAAATGCTATCAGCATGG-3′), forward primer 2 (5′-CTGAAATGCTATCAGCATGGCAAAGTGGTGACCTGCCATCGTGATGCG-3′), forward primer 3 (5′-CATAATACCGGCCTGCCGTTTCGTAATCTGAAACTGATTCTGCAG-3′), forward primer 4 (5′-TCCTCCTGCTCCGAAACCGAAAACAACAAATGCTGCTCCAC-3′), reverse primer 1 (5′-GGCAGGCCGGTATTATGATAGCAAAATTTCGCATCACGATGGCAGG-3′), reverse primer 2 (5′-GTTTCGGAGCAGGAGGAGGAGCAGCCCTGCAGAATCAGTTTCAG-3′), and reverse primer 3 containing the restriction enzyme XhoI site and a stop codon (5′-CTCTCGAGTCATTTATTGCAACGATCGGTGGAGCAGCATTTGTTG-3′). The target PCR fragment encoding the peptide gene was gel-purified, digested with EcoRI and XhoI, and cloned into the expression vector pET32b+ (Novagen, Merck KGaA, Darmstadt, Germany). The construct was confirmed by sequencing.

### 5.4. Recombinant Mamb-AL Production

Recombinant peptide production was achieved through fusion with a thioredoxin domain. *Escherichia coli* SHuffle cells transformed with the expression construct were cultured at 37 °C in LB medium containing 100 µg/mL of ampicillin until the culture optical density (OD_600_) reached 0.6–0.8. Protein expression was induced by adding 0.2 mM isopropyl β-D-1-thiogalactopyranoside (IPTG), and the cells were further cultured at 20 °C for 18 h. The cells were harvested, resuspended in the affinity chromatography start buffer (400 mM NaCl, 20 mM Tris-HCl, pH 7.5), ultrasonicated, and centrifuged at 14,000× *g* for 15 min to remove insoluble debris. The supernatant was applied to TALON Superflow metal affinity resin (Takara Bio USA, Inc., San Jose, CA, USA), and the fusion protein was purified following the manufacturer’s protocol. The eluate containing the fusion protein was incubated at 4 °C for 5–7 days. Protein cleavage was performed overnight at room temperature in the dark by adding HCl to a final concentration of 0.2 M and CNBr at a molar ratio of 600:1 (CNBr:fusion protein). Recombinant peptide was purified from the reaction mixture using reverse-phase high-performance liquid chromatography (HPLC) on a Jupiter C5 column (10 × 250 mm, Phenomenex, Torrance, CA, USA). Additional purification was performed on a Synergi Polar-RP column (4.6 × 250 mm, Phenomenex). The purity of the target product was assessed by RP-HPLC using an analytical Luna C18 column (2 × 100 mm, Phenomenex), followed by quantification based on the UV absorbance peak area at 220 nm, and confirmed by electrospray ionization mass spectrometry (ESI-MS).

### 5.5. Mass Spectrometry

Liquid chromatography–mass spectrometry (LC–MS) analysis of the peptide sample was performed on a Waters ACQUITY UPLC H-Class system (Waters Co., Milford, MA, USA). Samples were separated using an ACQUITY UPLC BEH C18 column (2.1 × 50 mm, Waters Co.) at a flow rate of 0.45 mL/min under a linear gradient of acetonitrile in water with 0.1% trifluoroacetic acid (TFA). Ultraviolet (UV) data were collected at 220 nm.

### 5.6. Expression of ASIC Channels in Xenopus laevisOocytes

Female *X.laevis* frogs were anesthetized with 0.17% ethyl 3-amino-benzoate methane sulfonate (MS222). A small ovarian section was surgically removed via an abdominal incision and placed in ND96 medium (96 mM NaCl, 2 mM KCl, 1.8 mM CaCl_2_, 1 mM MgCl_2_, and 5 mM HEPES, pH 7.4, adjusted with NaOH). Postoperative frogs displayed no signs of distress, and measures were taken to minimize suffering. Each animal underwent surgeries at intervals of no less than three months. Connective tissue and follicular sheaths were removed by incubating ovarian tissue in calcium-free ND96 medium containing collagenase (1 mg/mL) for 2 h at 21 °C. Healthy, mature stage IV and V oocytes were selected and stored in ND96 medium. PCi plasmids encoding rat ASIC1a, ASIC1b, or ASIC3 served as templates for synthesizing the corresponding messenger RNAs (mRNAs), which were then microinjected into oocytes (0.2–10 ng per oocyte) using a Nanolitre 2000 microinjection system (World Precision Instruments, Sarasota, FL, USA) 16–18 h after oocyte isolation. Injected oocytes were incubated in ND96 medium supplemented with penicillin and gentamicin (50 µg/mL each) at 18 °C for 2–3 days, followed by storage at 14 °C for up to 7 days.

### 5.7. Electrophysiological Assay

Whole-cell ASIC currents were measured using the two-electrode voltage clamp (TEVC) method with a GeneClamp 500 amplifier (Axon Instruments, Union City, NJ, USA). The holding potential maintained at –50 mV. Glass microelectrodes (0.5–2.5 MΩ) were filled with 3 M KCl solution, and ND96 (pH 7.4) was used as the external bath solution. ASIC currents were activated by rapidly switching the external solution from ND96 to either ND96 containing 10 mM MOPS (pH 6.7) or ND96 containing 10 mM MES (pH 5.5). Solution exchange was controlled using a custom-built computer-driven valve system. Data were filtered at 10 Hz and digitized at 100 Hz using an L780M ADC (LCard, Moscow, Russia). Peptides Mamb and Mamb-AL were applied for 30 s at pH 7.4 prior to acid stimulation of the channels.

### 5.8. Acetic Acid-Induced Writhing Pain Test

Adult (8–10 weeks) male ICR mice (25–30 g, Animal Breeding Facility Branch of the Shemyakin–Ovchinnikov Institute of Bioorganic Chemistry, Russian Academy of Sciences, Pushchino, Russia) were housed in Type 3 standard polycarbonate cages (820 cm^2^) with LIGNOCEL BK 8/15 bedding (JRS, Rosenberg, Germany) under controlled conditions (23 ± 2 °C, 12h light–dark cycle) with ad libitum access to food and water. Test substances were dissolved in saline and administered intramuscularly 90 min before testing. The visceral pain assay was performed as described by [49]. Distinct groups (n = 8 for each group) of mice were administered with 0.6% acetic acid in saline intraperitoneally (i.p.). Writhing behavior was recorded for 15 min after the mice were placed in transparent glass cylinders. A blinded observer assessed the responses. No formal power analysis was performed prior to the experiments, as the effect size and variability of the peptide treatment were unknown. However, because the initial results were highly significant, no additional animals were used. No animals were excluded from the analysis. Peptides (Mamb and Mamb-AL) were compared with a saline control group using analysis of variance (ANOVA) with post hoc Dunnett’s test. Data are presented as mean ± standard deviation (SD).

### 5.9. Data Analysis

Electrophysiological and behavioral data were analyzed using OriginPro 8.6.0 (OriginLab, Washington, DC, USA). Dose–response data for peptide inhibitory effects were fitted using the Hill equation: I_x_ = I_0_/[1 + ([x_0_]/[x])ˆn_H_], where I_x_ is the ionic current amplitude at a given peptide concentration [x], I_0_ is the current amplitude in the absence of peptide, x_0_ is the concentration at which the peptide exhibits half-maximal inhibition, and n_H_ is the Hill coefficient. The rate of current decay was fitted using a single exponential decay function: F(x) = A*eˆ(−x/τ_des_) + A_0_, where A_0_ is the minimal current response and τ_des_ is the decay constant. Statistical significance between the Mamb and Mamb-AL groups was determined using an unpaired *t*-test. Data are presented as mean ± SEM.

## Figures and Tables

**Figure 1 toxins-17-00101-f001:**
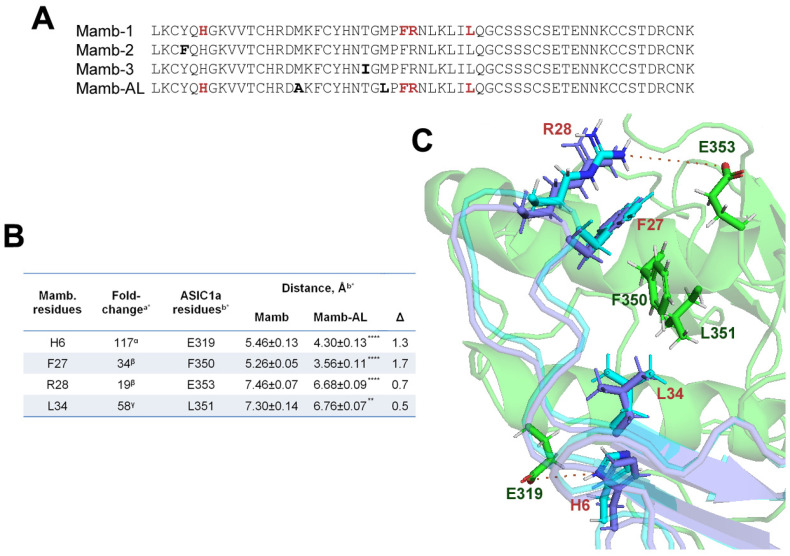
Molecular docking of wild-type Mamb peptides and their mutant analog Mamb-AL reveals tighter binding of Mamb-AL to the rat ASIC1a channel. (**A**) Primary structure of mambalgin peptides (Mamb-1, -2, and -3) and the mutant analog Mamb-AL. Residues differing from Mamb-1 are highlighted in bold, while residues critical for ASIC1a channel inhibition are marked in red. (**B**) Experimental and modeling data. ^a*^ shows literature data for Mamb-1, demonstrating its reduced activity (fold decrease) against the ASIC1a channel upon substitution of the residues listed in the first column with alanine. ^b*^ presents our modeling data, showing distances between the residues (first column) of Mamb and Mamb-AL and the channel (third column), along with the differences (Δ) between Mamb and Mamb-AL; ** *p*< 0.01, **** *p*< 0.001 compared with the corresponding Mamb residue, unpaired *t*-test. (**C**) Spatial model of interactions between wild-type Mamb (purple) and Mamb-AL (cyan) with the rat ASIC1a channel. Channel residues interacting with pharmacophore residues on the peptides (red) are shown in dark green. Distances are indicated by red dotted lines. References: α [23,24,26], β [23,24,31], γ [31].

**Figure 2 toxins-17-00101-f002:**
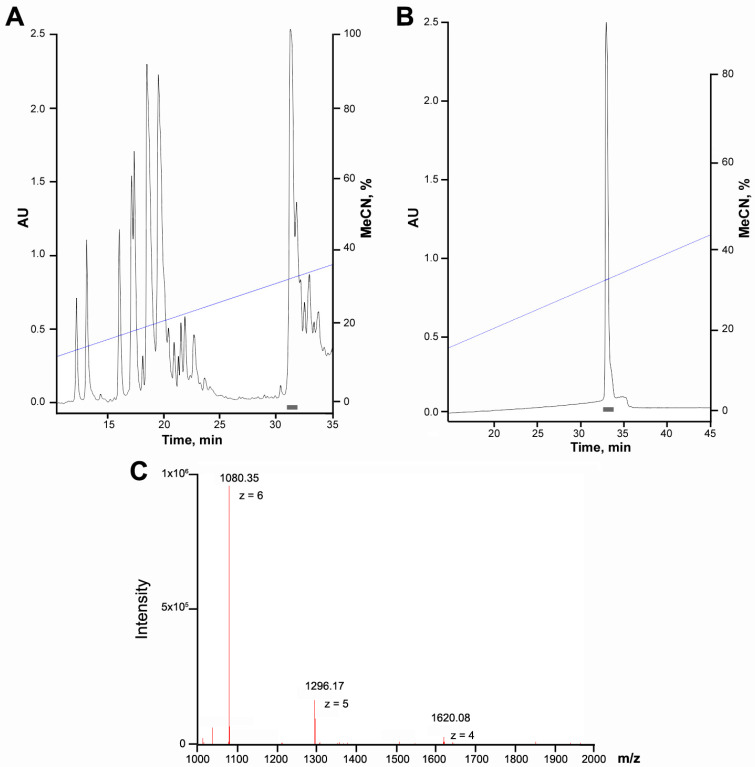
Purification of the target peptide Mamb-AL. (**A**) First stage of separation of the reaction mixture after cyanogen bromide hydrolysis of the chimeric protein, performed on a Jupiter C5 reverse-phase column (10 × 250 mm) in 0.1% trifluoroacetic acid (TFA) at a flow rate of 5 mL/min, using a linear gradient of acetonitrile (MeCN). (**B**) Second stage of separation on a Synergi Polar-RP column (4.6 × 250 mm) with a linear gradient of MeCN in 0.1% TFA at a flow rate of 1 mL/min. Fractions marked with a grey box correspond to the target product. Absorbance at 210 nm is shown as a black curve, and the MeCN gradient is represented by a blue line. (**C**) Mass spectrum of the Mamb-AL peptide, showing the average molecular weights of charged particles and their respective charges. The observed mass of 6476.12 Da closely matched the theoretical mass of 6476.42 Da.

**Figure 3 toxins-17-00101-f003:**
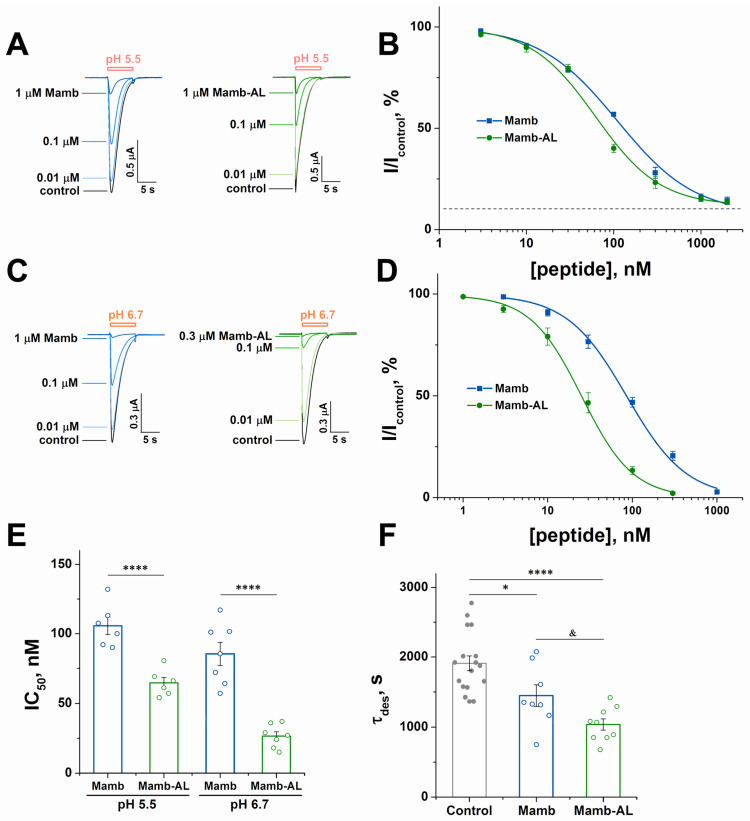
The inhibitory effect of Mamb-AL on rASIC1a exceeds that of Mamb. (**A**,**C**) Representative current traces showing the effects of various concentrations of Mamb (left panels) and Mamb-AL (right panels) on rASIC1a at pH stimuli of 5.5 (**A**) and 6.7 (**C**). Currents were induced by pH stimuli after a conditioning pH of 7.4, with peptides applied for 30 s prior to channel activation. The holding potential (Vh) was set to −50 mV, and the interval between applications varied from 1 to 2 min. (**B**,**D**) Dose–response curves for the inhibitory effects of Mamb and Mamb-AL on rASIC1a at pH stimuli of 5.5 (**B**) and 6.7 (**D**). Each data point represents measurements from six cells. (**E**) Bar plot showing statistical analysis of IC_50_ values calculated from the dose–response fits for the peptides’ inhibitory effects on individual cells (n = 6). **** *p*< 0.001; one-way ANOVA followed by Dunnett’s post hoc test. (**F**) Statistical analysis of the effects of 1 µM peptide concentration on the decay constant (τdes) at a pH stimulus of 5.5. Each data point represents data from 8 to 17 cells. * *p*< 0.05, **** *p*< 0.001 compared with control, paired *t*-test; & *p*< 0.05 comparing Mamb and Mamb-AL, unpaired *t*-test. Data are presented as mean ± S.E.M.

**Figure 4 toxins-17-00101-f004:**
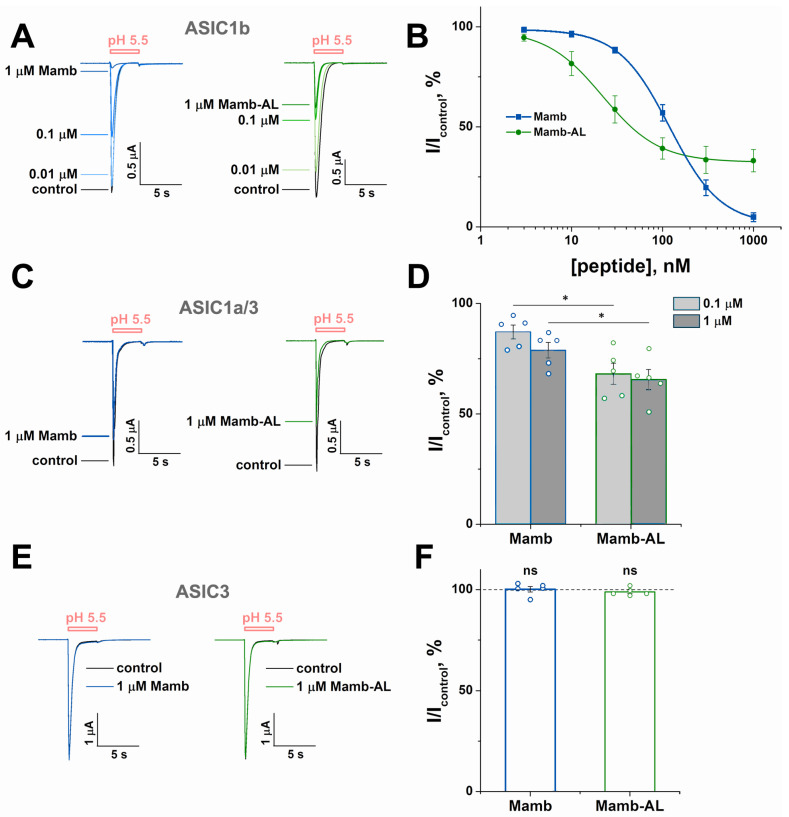
The inhibitory effect of Mamb-AL on other ASIC1-containing channels exceeds that of Mamb. (**A**,**C**) Representative current traces showing the effects of various concentrations of Mamb (left panels) and Mamb-AL (right panels) on homomeric rat ASIC1b (rASIC1b) (**A**) and heteromeric rat ASIC1a/3 (rASIC1a/3) channels (**C**). Currents were evoked by a pH 5.5 stimulus following a conditioning pH of 7.4, with peptides applied for 30 s prior to the pH stimulus. (**B**) Dose–response curves for the inhibitory effects of Mamb and Mamb-AL on rASIC1b. Each data point represents measurements from 5 to 7 cells. (**D**) Bar graph quantifying the inhibitory effects of Mamb and Mamb-AL on rASIC1a/3, shown as a percentage of the corresponding control currents (n = 5). Data are presented as mean ± SEM; * *p*< 0.05, unpaired *t*-test. (**E**,**F**) Representative current traces (**E**) and bar graph (**F**) demonstrating the absence of effects of Mamb and Mamb-AL on homomeric rat ASIC3 channels (n = 5). Data are shown as mean ± SEM; ns indicates no significant difference vs. control (dotted horizontal line), paired *t*-test.

**Figure 5 toxins-17-00101-f005:**
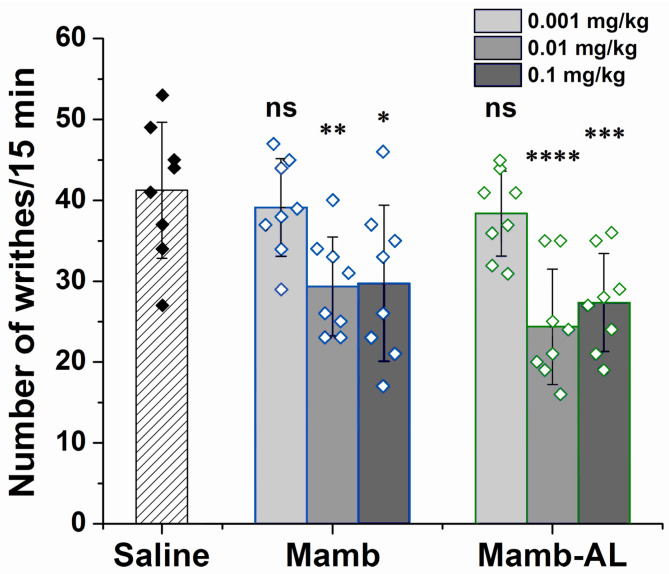
Analgesic effect of peptides Mamb and Mamb-AL in an acute pain model. Peptides were administered intramuscularly 90 min before testing. Their efficacy was assessed using the acetic acid-induced writhing test, where the number of writhes was counted after intraperitoneal administration of 0.6% acetic acid. Results are presented as mean ± SD (n = 8 for each group). Statistical analysis was performed using one-way ANOVA followed by Dunnett’s post hoc test. * *p*< 0.05, ** *p*< 0.01, *** *p*< 0.005, **** *p*< 0.001 indicate significant differences compared with the control group, and ns indicates no significant difference.

## Data Availability

The original contributions presented in this study are included in the article. Further inquiries can be directed to the corresponding author.

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
