# Peer review of "Two Amino Acid Substitutions Improve the Pharmacological Profile of the Snake Venom Peptide Mambalgin"

_toxins, 2025, doi:10.3390/toxins17030101_

Round 1

Reviewer 1 Report

Comments and Suggestions for Authors

In the present manuscript, Mamb-AL, a mutant analog of mambalgin-1, has been developed to optimise recombinant peptide production and enhance pharmacological properties. Electrophysiological experiments and a mouse model of acid-induced pain demonstrate Mamb-AL's improved efficiency and analgesic efficacy. The study is very interesting and well-written. My main concerns are that the journal guidelines require results and discussion to be in separate sections. I also believe the authors need to enrich the discussion section with more comparisons to the existing literature. Additionally, I am curious about the toxicity of the three-finger toxins and their analogues.

1. Line 11. What are the complex steps? Why are they complex?  What does several mean? How many steps?

2. Line 14-15. Please specify the mouse model

3.  Some sentences are missing references. Please check lines 26-27, 27-28, 31-33, 33-35, 44-45, 49-50, 75-76

4. Lines 42. What are the various diseases? Please specify.

5.  Lines 76-79. What are the previous studies?

6. Line 79-81. The objectives should be clearly established in the introduction section, and not in the results section.

7. Figure 1. The amino acid residues highlighted in red are hard to see.

8. The physicochemical properties for the parent peptide and analogue can be estimated and compared. Do the substitutions induce significant changes in physicochemical parameters?

9. Figure 1B and C must be mentioned in the manuscript before the figure.

10. Lines 103-104. Do the authors mean figure 1?

11. Lines 107-109. Please provide the evidence supporting this statement. What were the parameters compared? Do the authors performed statistical test to confirm if the differences are significant? The values provide in the table shown in figure 1B seems to be very close. Please enrich the results sections and also the discussion around these findings.

12. Line 111. Please include references. Previous studies? What are they? I would like to read them. What are the challenges? The authors mentioned two in the following sentence, but they said several.

13. The title of the figures must highlight the main finding and not the methodology behind. The description in the figure legend should detail the methodology or approach.

14. The titles of results sections are focused on the methodology and not the results. I suggest revising with emphasis on the main result.

15.  Figure 2A, B and C should be clearly mentioned in the manuscript.

16. What was the purity level? This should be included in the results.

17. Figure 2. The axis should end in a number. There are lines above the last number.

18. Lines 136-139. Please avoid one sentence paragraph. Please combine similar ideas or expand the topic.

19. Figure 3C was mentioned before figure 3B. The authors should organise figure panel according to the text, to enhance comprehension.

20. Is the IC50 significantly different? A statistical test is relevant in this context.

21. The colours of graphs must be standardised. Different colours were used for the same conditions.

22. Please check the journal’s guidelines. Results and discussion are presented in different sections. On the other hand, the discussion must be enriched with more comparison with the literature.

23. Three-finger toxins are typically highly toxic. The authors have not addressed this toxicity. These toxins can cause neurotoxicity and, due to their membrane-disrupting properties, red blood cell lysis. Therefore, it is important to assess the side effects, such as evaluating the toxicity to red blood cells. Haemolytic assays are straightforward but may require samples, which could be a limitation for the authors. However, the authors can also use AI-driven tools to predict haemolytic properties. Several web servers are available that can predict this activity in seconds using the sequences. This review article (PMCID: PMC8953747) summarises various freely available tools. I suggest the authors utilise these tools and use this manuscript or others addressing toxicity to enrich the discussion section. The selectivity and toxicity assessment are valuable for drug discovery purposes.

24. The conclusion sections can be summarised emphasising the main findings.

25. Line 242. Please the author’s name.

26. The authors must include the methodology to assess the level of purity. For example, peak integration area and mass spectrometry, which unveil the molecular identity. Although the purity seems to be very high, the chromatogram shows small impurities close to the peak.

27. The format of the references must be checked. In some cases, all words are capitalised, while in others they are not.

28. Limitations and perspectives of the study were not included in the current version of the manuscript. 

Author Response

Comments 1: In the present manuscript, Mamb-AL, a mutant analog of mambalgin-1, has been developed to optimise recombinant peptide production and enhance pharmacological properties. Electrophysiological experiments and a mouse model of acid-induced pain demonstrate Mamb-AL's improved efficiency and analgesic efficacy. The study is very interesting and well-written.

Response 1: We appreciate the positive, insightful, and valuable evaluation of our work.

Comments 2: My main concerns are that the journal guidelines require results and discussion to be in separate sections. I also believe the authors need to enrich the discussion section with more comparisons to the existing literature. Additionally, I am curious about the toxicity of the three-finger toxins and their analogues.

Response 2: We have added a Discussion section following your recommendations, where we provide more comparisons with known data and describe the absence of toxicity in these representatives of three-finger toxins.

Comments 3: Line 11. What are the complex steps? Why are they complex?  What does several mean? How many steps?

Response 3: By "complex steps," we meant additional stages in the target product purification scheme that make the process more complicated, such as inclusion body isolation, refolding, dialysis, ion exchange and size-exclusion chromatography. To avoid overcomplicating and cluttering the Abstract with details, we removed the phrase "by eliminating several complex steps" and instead provided these specifics in the Discussion section.

Comments 4: Line 14-15. Please specify the mouse model.

Response 4: We specified the model as an acetic acid-induced writhing pain model.

Comments 5: Some sentences are missing references. Please check lines 26-27, 27-28, 31-33, 33-35, 44-45, 49-50, 75-76.

Response 5: Missing references were added.

Comments 6: Lines 42. What are the various diseases? Please specify.

Response 6: We specified the diseases and added the necessary references.

Comments 7: Lines 76-79. What are the previous studies?

Response 7: We moved this phrase to the Discussion section and added the necessary references.

Comments 8: Line 79-81. The objectives should be clearly established in the introduction section, and not in the results section.

Response 8: We removed this phrase from the Results.

Comments 9: Figure 1. The amino acid residues highlighted in red are hard to see.

Response 9: We fixed it.

Comments 10: The physicochemical properties for the parent peptide and analogue can be estimated and compared. Do the substitutions induce significant changes in physicochemical parameters?

Response 10: We estimated and compared the physicochemical properties of the parent peptide and its analog. This analysis was added to the Results, Discussion, and Table A1, where we show that key physicochemical parameters, such as isoelectric point, extinction coefficient, and hydrophobicity, remain unchanged.

Comments 11: Figure 1B and C must be mentioned in the manuscript before the figure.

Response 11: We fixed it.

Comments 12: Lines 103-104. Do the authors mean figure 1?

Response 12: Yes, we meant Figure 1. Corrections were made.

Comments 13: Lines 107-109. Please provide the evidence supporting this statement. What were the parameters compared? Do the authors performed statistical test to confirm if the differences are significant? The values provide in the table shown in figure 1B seems to be very close. Please enrich the results sections and also the discussion around these findings.

Response 13: Statistical analysis was performed for the five models with the lowest interaction scores and RMSD parameters. Table A2 presents the Rosetta scores, showing that the peptides did not differ significantly based on these parameters. Therefore, we chose to compare the peptides based on the distances between key pharmacophores of the peptide and their interacting residues on the channel. The revised Table 2B provides the mean values of these distances for the selected five models, along with the results of the statistical analysis, which demonstrate that these differences are statistically significant.

Comments 14: Line 111. Please include references. Previous studies? What are they? I would like to read them. What are the challenges? The authors mentioned two in the following sentence, but they said several.

Response 14: We have revised this sentence as follows: "Previous studies faced significant challenges in obtaining sufficient amounts of Mamb peptides due to several factors". Additionally, we decided to expand on this statement by listing the main challenges and moving it to the Discussion section. We have also included all necessary references. The key challenges, in our opinion, are as follows: a) the severe limitation of the natural source (i.e., snake venom); b) the relatively large size of the peptide (57 amino acids), which makes its production via chemical synthesis followed by refolding expensive; c) the low yields of recombinant analogs in bacterial expression systems, requiring purification from the periplasmic space; d) the complex procedure for isolating the recombinant analog from inclusion bodies in bacterial systems.

Comments 15: The title of the figures must highlight the main finding and not the methodology behind. The description in the figure legend should detail the methodology or approach.

Response 15: We corrected the title of the figures according to your recommendations.

Comments 16: The titles of results sections are focused on the methodology and not the results. I suggest revising with emphasis on the main result.

Response 16: We corrected the titles of results sections according to your recommendations.

Comments 17: Figure 2A, B and C should be clearly mentioned in the manuscript.

Response 17: We fixed it.

Comments 18: What was the purity level? This should be included in the results.

Response 18: We included the purity level in the Results and Figure A1.

Comments 19: Figure 2. The axis should end in a number. There are lines above the last number.

Response 19: We fixed it.

Comments 20: Lines 136-139. Please avoid one sentence paragraph. Please combine similar ideas or expand the topic.

Response 20: We expanded the topic.

Comments 21: Figure 3C was mentioned before figure 3B. The authors should organise figure panel according to the text, to enhance comprehension.

Response 21: We reorganized the figure panel according to your recommendations.

Comments 22: Is the IC50 significantly different? A statistical test is relevant in this context.

Response 22: We performed a comparison using a one-way analysis of variance followed by a Dunnett's test. The obtained p-values for IC50 at pH 5.5 and pH 6.7 were 0.000256 and 0.0000262, respectively. We have corrected the name of the statistical test in the caption for Figure 3.

Comments 23: The colours of graphs must be standardised. Different colours were used for the same conditions.

Response 23: We fixed it.

Comments 24: Please check the journal’s guidelines. Results and discussion are presented in different sections. On the other hand, the discussion must be enriched with more comparison with the literature.

Response 24: We added the Discussion section.

Comments 25: Three-finger toxins are typically highly toxic. The authors have not addressed this toxicity. These toxins can cause neurotoxicity and, due to their membrane-disrupting properties, red blood cell lysis. Therefore, it is important to assess the side effects, such as evaluating the toxicity to red blood cells. Haemolytic assays are straightforward but may require samples, which could be a limitation for the authors. However, the authors can also use AI-driven tools to predict haemolytic properties. Several web servers are available that can predict this activity in seconds using the sequences. This review article (PMCID: PMC8953747) summarises various freely available tools. I suggest the authors utilise these tools and use this manuscript or others addressing toxicity to enrich the discussion section. The selectivity and toxicity assessment are valuable for drug discovery purposes.

Response 25: In the added Discussion section, we provide literature examples demonstrating the absence of toxicity in mambalgins. Additionally, following your recommendations, we conducted an in silico analysis using the HemoPI (Hemolytic Activity Prediction) program, which showed that the PROB scores for peptide fragments scanned from Mamb and Mamb-AL ranged from 0.47 to 0.51. A score of 1 indicates a high likelihood of hemolytic activity, while a score of 0 suggests a very low likelihood.

Comments 26: The conclusion sections can be summarised emphasising the main findings.

Response 26: We corrected the Conclusion section according to your recommendations.

Comments 27: Line 242. Please the author’s name.

Response 27: The author's name was added.

Comments 28: The authors must include the methodology to assess the level of purity. For example, peak integration area and mass spectrometry, which unveil the molecular identity. Although the purity seems to be very high, the chromatogram shows small impurities close to the peak.

Response 28: We have added a description of the target peptide purity assessment in the Materials and Methods section (subsection 5.4), along with Figure A1.

Comments 29: The format of the references must be checked. In some cases, all words are capitalised, while in others they are not.

Response 29: We fixed it.

Comments 30: Limitations and perspectives of the study were not included in the current version of the manuscript.

Response 30: We have made an effort to describe the limitations and perspectives of our study in the Discussion and Conclusion sections.

Reviewer 2 Report

Comments and Suggestions for Authors

This paper simplifies the purification process and improves the expression yield by mutating two amino acids (methionine to alanine and methionine to leucine substitutions) of mambalgin-1 to eliminate methionine from the peptide sequence, resulting in the use of cytochrome bromamide to cleave the fusion protein. The activity of peptides expressed simultaneously has been increased. This paper has certain significance for improving the preparation process of mambalgin. However, there are still some questions on the paper:

  1. In the structural simulation, the author only mentioned the shortening of the distance between some interacting amino acids in the simulation. However, shortening the distance may not necessarily mean an increase in activity. The author should supplement and discuss other parameters related to affinity, such as changes in free energy. The author used RosettaDock 5.0 for simulation. Can other simulation methods be added, such as AlfaFold?
  2. The position of mambalgin-1 should be marked in Figure 2A, which can be indicated by arrows or described in the main text for retention time. How are expressed proteins detected during the purification process? Is there SDS-PAGE available?
  3. In Figure 4C, the Dose Response curves of mambalgin-1 need to be verified and discussed. After peptide mutation, the inhibitory activity of ASIC is enhanced, indicating that the mutation may have increased the affinity between the peptide and ASIC, allowing it to exert inhibitory effects at lower concentrations; However, due to changes in the binding site, it is unable to completely block the activity of the channel like wild-type peptides, resulting in a decrease in maximum inhibition rate. Peptide mutations may affect its own conformation, thereby altering its interaction with ASIC channels. The Hill coefficient reflects synergy, and weakening indicates that the binding between the mutated peptide and ASIC may change from positive synergy to non-synergy or negative synergy, suggesting a change in binding site or conformation that affects the interaction between the peptide and the channel.
  4. In the analgesic activity assay of Figure 5, The activity of mamgalgin-1 at 0.1 mg/Kg and 0.01 mg/Kg is comparable, and there is even a trend of decreased activity. The author should discuss the reasons. In addition, animals were used in the experiment, and relevant ethical approval documents or explanations should be supplemented. Meanwhile, the experiment lacks a positive control drug for pain relief, such as morphine.

Author Response

Comments 1: This paper simplifies the purification process and improves the expression yield by mutating two amino acids (methionine to alanine and methionine to leucine substitutions) of mambalgin-1 to eliminate methionine from the peptide sequence, resulting in the use of cytochrome bromamide to cleave the fusion protein. The activity of peptides expressed simultaneously has been increased. This paper has certain significance for improving the preparation process of mambalgin.

Response 1: We appreciate the positive and valuable evaluation of our work.

Comments 2: However, there are still some questions on the paper: In the structural simulation, the author only mentioned the shortening of the distance between some interacting amino acids in the simulation. However, shortening the distance may not necessarily mean an increase in activity. The author should supplement and discuss other parameters related to affinity, such as changes in free energy. The author used RosettaDock 5.0 for simulation. Can other simulation methods be added, such as AlfaFold?

Response 2: 

As a result, we selected five models with the lowest energy parameters and RMSD values. These models exhibited no significant differences in terms of total energy and interaction scores, likely because, in addition to the identified pharmacophore residues, other amino acid residues also contribute to these values. Therefore, we decided to analyze the distances between the peptide pharmacophore residues and the channel. A detailed examination revealed that, for the Mamb-AL peptide, these distances were smaller, suggesting potentially stronger interactions. We have added the values of total energy and interaction scores (Table A2), as well as the corresponding section in the Discussion.

We did not use AlphaFold for docking, as its capabilities in this area are more limited compared to Rosetta. This limitation arises because AlphaFold relies on machine learning trained on known molecular structures and their conformational states. In contrast, Rosetta simulates molecular interactions based on physical principles and laws, employing the Monte Carlo algorithm to explore conformational space more extensively.

Comments 3: The position of mambalgin-1 should be marked in Figure 2A, which can be indicated by arrows or described in the main text for retention time. How are expressed proteins detected during the purification process? Is there SDS-PAGE available?

Response 3: We provided a comparison of peptide retention times in the Results section (subsection 2.2). Detection of the target product was performed based on UV absorption spectra at wavelengths of 210, 220, and 280 nm during chromatographic separations. Additionally, the target peptide was confirmed by mass spectrometry. Since the target protein was expressed in significant quantities and could be easily detected during chromatographic separation, we believe that performing electrophoretic separation of proteins under denaturing conditions was redundant in this case.

Comments 4: In Figure 4C, the Dose Response curves of mambalgin-1 need to be verified and discussed. After peptide mutation, the inhibitory activity of ASIC is enhanced, indicating that the mutation may have increased the affinity between the peptide and ASIC, allowing it to exert inhibitory effects at lower concentrations; However, due to changes in the binding site, it is unable to completely block the activity of the channel like wild-type peptides, resulting in a decrease in maximum inhibition rate. Peptide mutations may affect its own conformation, thereby altering its interaction with ASIC channels. The Hill coefficient reflects synergy, and weakening indicates that the binding between the mutated peptide and ASIC may change from positive synergy to non-synergy or negative synergy, suggesting a change in binding site or conformation that affects the interaction between the peptide and the channel.

Response 4: Thank you for your insightful comment. We have included a discussion on the effects of the mutant peptide and the wild-type peptide on the ASIC1b isoform in the Discussion section.

Comments 5: In the analgesic activity assay of Figure 5, The activity of mamgalgin-1 at 0.1 mg/Kg and 0.01 mg/Kg is comparable, and there is even a trend of decreased activity. The author should discuss the reasons. In addition, animals were used in the experiment, and relevant ethical approval documents or explanations should be supplemented. Meanwhile, the experiment lacks a positive control drug for pain relief, such as morphine.

Response 5: We agree that a trend of decreased activity is observed between the 0.1 and 0.01 mg/kg doses, although the difference between them is not statistically significant. The possible reasons for this phenomenon are discussed in the Discussion section, where we provide examples showing that hormesis is a common occurrence among biologically active compounds with pharmacological effects, including negative modulators of ASIC channels.

We have provided the relevant ethical approval documents or explanations; however, this information was redacted by the editors due to the double-blind peer review process.

Unfortunately, we are unable to use morphine as a positive control drug, as we do not have the necessary permits to work with opioids and related substances.

Round 2

Reviewer 1 Report

Comments and Suggestions for Authors

All major points have been addressed. The manuscript appears to be in good shape for publication.

Reviewer 2 Report

Comments and Suggestions for Authors

The authors have answered all the questions and the quality of the manuscript has been greatly improved. I have no more comments.